# Monitoring of Plant Ecological Units Cover Dynamics in a Semiarid Landscape from Past to Future Using Multi-Layer Perceptron and Markov Chain Model

Masoumeh Aghababaei [1], Ataollah Ebrahimi [1,*], Ali Asghar Naghipour [1], Esmaeil Asadi [1] and Jochem Verrelst [2]

1 Department of Nature Engineering, Faculty of Natural Resources and Earth Sciences, Shahrekord University, Shahrekord 8818634141, Iran; ma.aghababaeei@stu.sku.ac.ir (M.A.); aa.naghipour@sku.ac.ir (A.A.N.); asadi-es@sku.ac.ir (E.A.)
2 Image Processing Laboratory (IPL), Parc Científic, Universitat de València, 46980 Paterna, Spain; jochem.verrelst@uv.es
* Correspondence: ataollah.ebrahimi@sku.ac.ir; Tel.: +98-9132808343

**Abstract:** Anthropogenic activities and natural disturbances cause changes in natural ecosystems, leading to altered Plant Ecological Units (PEUs). Despite a long history of land use and land cover change detection, the creation of change detection maps of PEUs remains problematic, especially in arid and semiarid landscape. This study aimed to determine and describe the changes in PEUs patterns in the past and present, and also predict and monitor future PEUs dynamics using the multi-layer perceptron-Markov chain (MLP-MC) model in a semiarid landscape in Central Zagros, Iran. Analysis of PEUs classification maps formed the basis for the identification of the main drivers in PEUs changes. First, an optimal time-series dataset of Landsat images were selected to derive PEUs classification maps in three periods, each separated by 16 years. Then, PEUs multi-temporal maps classified for period 1 (years 1986–1988) period 2 (years 2002–2004), and period 3 (years 2018–2020) were employed to analyze and predict PEUs dynamics. The dominant transitions were identified, and the transition potential was determined by developing twelve sub-models in the final change prediction process. Transitions were modeled using a Multi-Layer Perceptron (MLP) algorithm. To predict the PEU map for period 3, two PEUs classification maps of period 1 and period 2 were used using the MLP-MC method. The classified map and the predicted map of period 3 were used to evaluate and validate the predicted results. Finally, based on the results, transitions of future PEUs were predicted for the year 2036. The MLP-MC model proved to be a powerful model that can predict future PEUs dynamics that are the result of current human and managerial activities. The findings of this study demonstrate that the impact of anthropogenic processes and management activities will become visible in the natural environment and ecosystem in less than a decade.

**Keywords:** plant ecological unit's changes; land change modeler; time-series dataset; Markov chain model

## 1. Introduction

Land cover is the combination of biotic and abiotic physical substances on the earth's surface. Typically, it includes natural vegetation (forests, grassland), water, soil, and man-made surfaces [1,2]. Mapping and monitoring changes in land cover types are pivotal for land cover sustainability planning and environmental management needs [3–5]. However, the benefits and importance of land cover maps based on plant ecological units (PEUs) are less understood [6]. PEUs represent the potential plant communities that can occur on a site. Because of differences in elevation, soil, historic background, and land abandonment processes, a variety of plant communities with distinctive amounts and types of vegetation can be found in an area. Thereby, the identification of PEUs provides a reference for the

interpretation of land cover data and research, monitoring, and land management [7]. PEUs mapping through classification techniques is extremely important, and their accuracy will directly affect the extraction of the final prediction results and change detection maps [8]. Therefore, documenting and evaluating the temporal dynamics of PEUs and making predictions of plausible future PEUs dynamics is vital for plant community and land cover monitoring.

Optical remote sensing data offer a valuable source of information for studying change detection. Several studies in change detection have been related to land use/cover changes [9,10]. The key areas in these land use studies included urban areas [11,12], drylands [13], protected areas [14], and wetlands [8,15]. Modeling PEUs changes, on the other hand, not only helps evaluate the current condition of land cover but also helps managers and natural resource planners prevent or reduce negative consequences of undesirable future PEUs changes. Meanwhile, PEUs are hardly distinguishable due to their similar spectral behavior and low inter-class separability, especially in arid and semiarid areas. Therefore, these PEUs impose challenges to classification and predictions of future changes. In a related study about improving land cover map accuracy, Aghababaei et al. [7] found that the development of an accurate land cover map is feasible in a semiarid rangeland when a dataset of time-series images is entered into the classification. So far, publicly available satellite data have been used in change detection studies, such as MODIS [16], SPOT [3], and Landsat [17,18]. Nevertheless, the most notable applications of Landsat data are land cover change analyses. Landsat is widely used in these studies due to its long-range time-series data and free access [19]. By exploiting the temporal dimension of Landsat data, the effects of poor-quality observations can be minimized in the classification and change detection [20]. Regarding the processing of multi-temporal images, the Google Earth Engine (GEE) platform allows researchers to classify and process large volumes of satellite images, including Landsat [21–23]. The combination of an unprecedented data catalog with powerful processing possibilities caused a paradigm shift in Optical Earth observation (EO) data analysis that moved away from traditional image analysis using desktop software to cloud-based processing.

Various models have been used in predicting LULC changes, such as the Markov Chain model (MC), Cellular Automata (CA), and Multi-Layer Perceptron–Markov Chain (MLP-MC). However, integrated models such as MLP-MC are implemented in the integration of the Multi-Layer Perceptron Neural Network (NN-MLP) model with the MC model as an effective approach in prediction studies [8]. This model is used to predict geospatial changes with the use of previous changes [24]. Studies show that the MC model gives better results when predicting long-term temporal LULC variations, future scenarios, and landscape changes [15,25–27].

Altogether, the objectives of the study presented here are the following. Firstly, to develop a classification strategy using a multi-temporal dataset that achieves accurate and efficient PEUs mapping. Secondly, using the MLP-MC to monitor changes in PEUs dynamics from 1986 to 2020 with 16-year time intervals and forecasting future changes for 16 years later, i.e., 2036, in a semiarid rangeland landscape in Southwest Iran, where the low vegetation covers make it a challenging task. This work can help natural resource managers and planners make a sustainable result for natural resource protection, climate regulation, and erosion control by understanding current and future PEUs changes.

## 2. Materials and Methods

### 2.1. Study Area

The study presented here is based on the Marjan watershed case study area located in the Chaharmahal-Va-Bakhtiari province in Southwest Iran. As shown in Figure 1, PEUs boundaries can be straightforwardly observed in this area due to the relatively narrow ecotones and sharp borders between them. The study area is characterized by a heterogeneous, semiarid landscape with a predominance of different PEUs. The covered area is 7736.58 ha. There are some effects of anthropogenic processes and management

activities present, leading to spatial changes in PEUs maps in different periods. We are not aware of work focusing on modeling these PEUs dynamics.

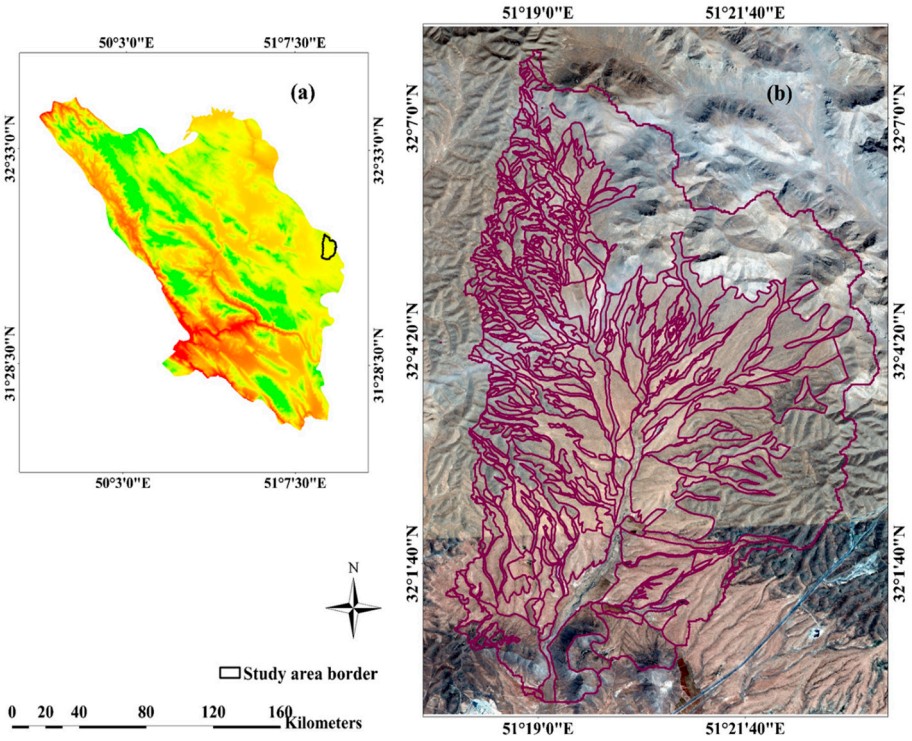

**Figure 1.** Location of Marjan in the Chaharmahal-Va-Bakhtiari province: (**a**) Chaharmahal-Va-Bakhtiari border; (**b**) study area border (Marjan).

### 2.2. Field Measurements

Four PEUs groups were distinguished in the area, including (1) PEU1 (*Astragalus verus Olivier* (As ve)), (2) PEU2 (*Bromus tomentellus Boiss* (Br to)), (3) PEU3 (*Scariola orientalis Sojak* (Sc or)), and (4) PEU4 (*Astragalus verus Olivier—Bromus tomentellus Boiss* (As ve—Br to)). Canopy cover data could potentially be used to identify structural and compositional PEUs or a combination of both, the so-called physiognomic–floristic classification, to gain a sound and accurate perspective on PEUs. We sampled the four identified PEUs using three replicates, in each of which canopy cover was sampled along three transects of 100 m that were evenly distributed throughout the study area. The sampling was randomly systematic (the first node was selected randomly but the rest were systematically distributed along the transects). We collected species-based canopy cover within each quadrat. In each PEUs, the canopy cover percentage was calculated and the PEUs were named according to their dominant floristic composition (Figure 2). For this purpose, first, the dominant plant species of each PEU was identified and then its accompanying species was determined, with the dominant species having 50% or more canopy cover than the previously dominant species. Thus, each PEU was named based on a physiognomic–floristic method.

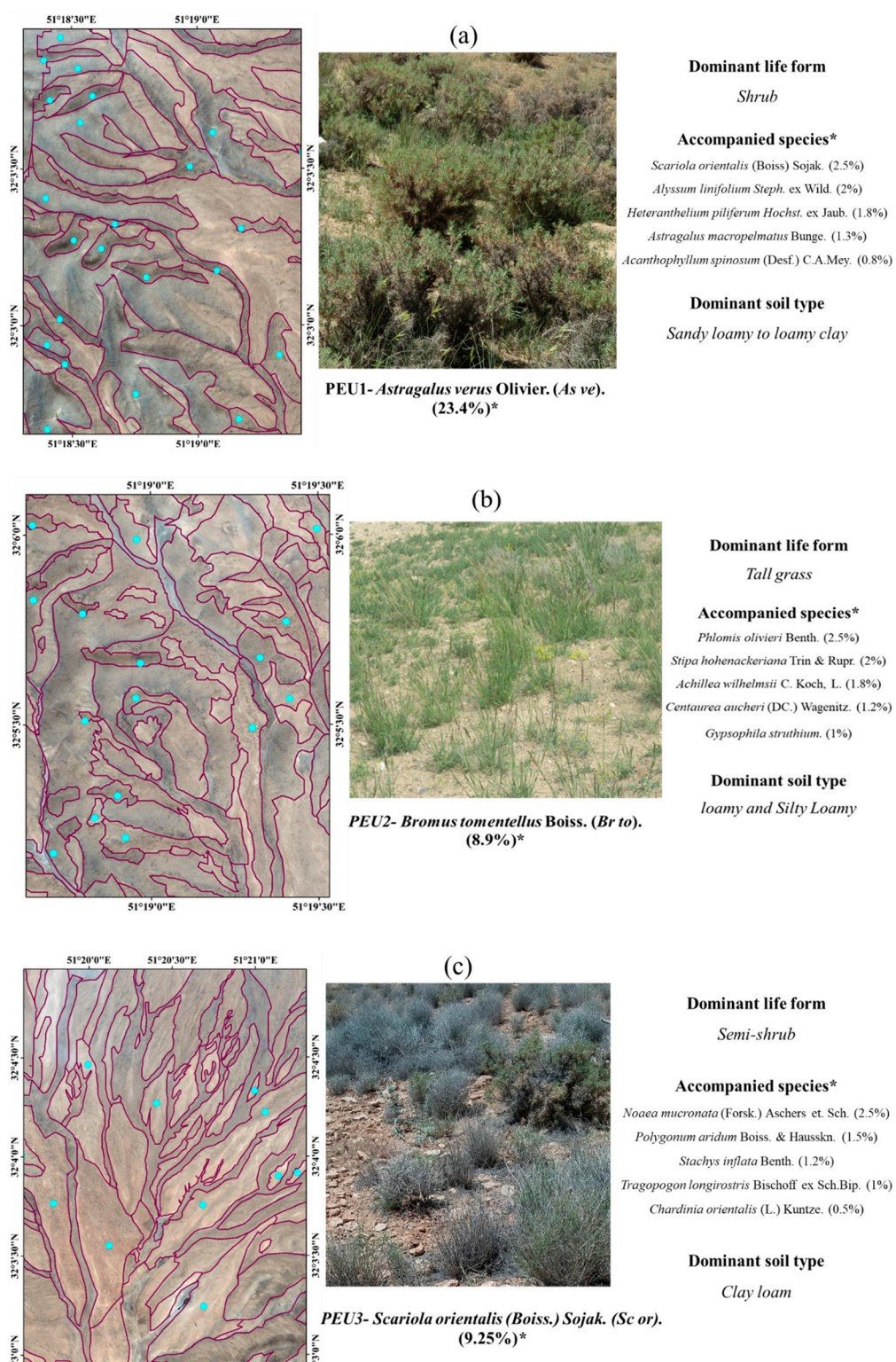

**Figure 2.** *Cont.*

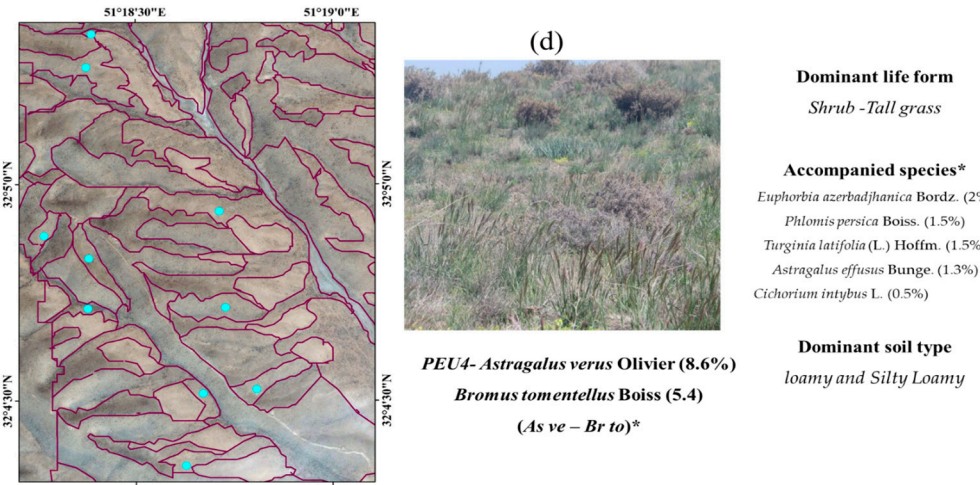

**Figure 2.** The location of PEUs in Google Earth images and the corresponding field photos. (**a**) PEU1 (As ve); (**b**) PEU2 (Br to); (**c**) PEU3 (Sc or); and (**d**) PEU4 (As ve—Br to). * Canopy cover percentage of dominant and accompanied species that was calculated on transects.

### 2.3. NDVI Spectral Curve and Selection of Optimal Time-Series Datasets

This study used multi-temporal satellite data. The satellite data consisted of 3 periods with 16-year time intervals acquired by Landsat 5 TM for period 1 (1986, 1987, 1988), Landsat 7 ETM+ for period 2 (2002, 2003, 2004), and Landsat 8 OLI for period 3 (2018, 2019, 2020). With the support of Google Earth Engine (GEE), we used the Top of Atmosphere (TOA) reflectance (ee. Image Collection ('LANDSAT/LT05/C01/T1_TOA')), (ee. Image Collection ('LANDSAT/LE07/C01/T1_TOA')), and (ee. Image Collection ('LANDSAT/LC08/C01/T1_TOA')) for Landsat 5, Landsat 7, and Landsat 8 images, respectively. Likewise, using the cloud filter (ee. Filter. Less than ('CLOUD_COVER', 5)), only satellite images with less than 5% cloud cover were selected. So, some images were filtered due to persistent cloudiness. Finally, the NDVI time-series profile was calculated from January to December of each year (Equation (1)).

By analyzing the NDVI temporal profile, we identified a multi-temporal dataset as an input for PEUs classification for each period [6].

$$NDVI = \frac{(NIR - Red)}{(NIR + Red)} \tag{1}$$

where *RED* is the reflectance in the red band, and *NIR* is the reflectance in the near-infrared band [24].

### 2.4. Methodology

Figure 3 shows the conducted workflow of PEUs change detection and prediction using the MLP-MC model. First, using the physiognomic–floristic method, we selected 4 dominant PEUs in the region, and using GPS, training and testing samples were collected by field survey in the study area. We used the RF classification algorithm and an optimal combination of multi-temporal images to produce PEUs maps for period 1 (years 1986–1988), period 2 (years 2002–2004), and period 3 (years 2018–2020), respectively. Detection of PEUs changes was based on the Markov Chain model (MLP-MC model) that was obtained from change analysis in PEUs classes from two dates.

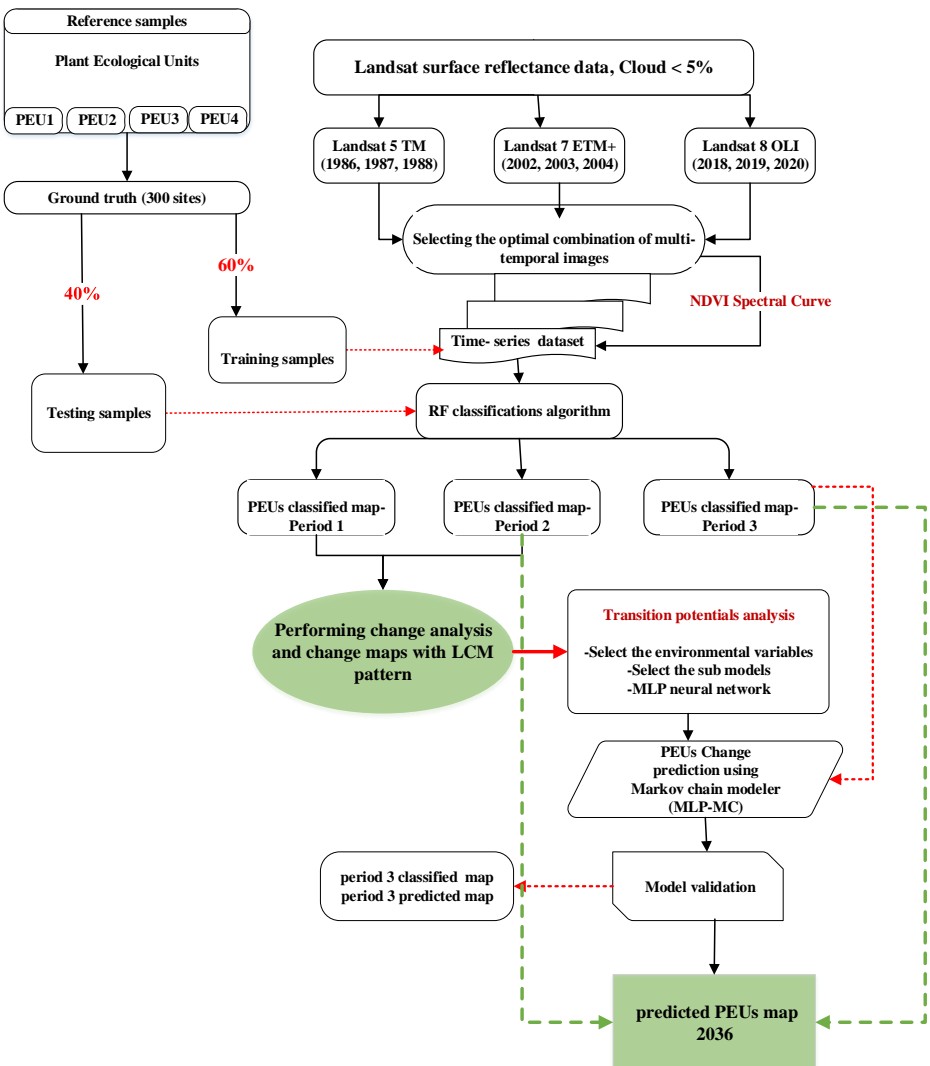

**Figure 3.** The workflow for PEUs change detection (1986–2020) and future prediction of PEUs (2020–2036) in this study.

For period 3, a prediction map was created using the forecast model and the PEUs classification maps for period 1 and period 2. To validate the accuracy of the predicted map, we used the PEU classification map of the same period (period 3 classified map and period 3 predicted map). Finally, using the MLP-MC model, we obtained a PEU prediction map for the year 2036.

### 2.4.1. Training and Verification Points

After identifying the dominant PEUs in the study area, in total, 300 points were recorded for the PEUs using a Garmin eTrex 32× GPS. Then, by examining aerial photographs from each period, it was ensured that each sample point was selected in the PEUs center. Finally, the points were divided into two groups, the "training samples" (60%) and "verification samples" (40%), used for the classification and validation of results.

### 2.4.2. PEUs Multi-Temporal Classification and Maps Validation

We used random forests (RF) to produce PEU maps for period 1 (years 1986–1988), period 2 (years 2002–2004), and period 3 (years 2018–2020). After selecting the multi-temporal dataset for each period, RF algorithm, as one of the most accurate machine learning algorithms [28–31], was used to perform PEUs classification. The mapping accuracy was evaluated by means of a confusion matrix (including Overall Accuracy (OA), Overall Kappa

(OK), User's Accuracy (UA), Producer's Accuracy (PA), and Kappa Index of Agreement (KIA)) for all years [32].

### 2.4.3. PEUs Change Detection and Analysis Using the LCM Model

Change detection and prediction mapping were achieved by the Land Change Modeler (LCM). The LCM consists of a suite of tools for evaluating losses and gains and assessment of land cover changes and transformations between the land cover classes in the map [33]. The PEUs maps based on the RF classification algorithm and an optimal combination of multi-temporal images of Landsat 5 TM, Landsat 7 ETM+, and Landsat 8 OLI images were used in order to quantify the PEUs. The recognition of changes can facilitate the determination of future PEUs changes and scenarios. The detection of PEUs changes are based on the Markov Chain matrix that is obtained from change analysis in PEUs classes from two dates (period 1 to period 2). In addition, this matrix can predict future PEUs changes [26]. Based on the detection and analysis of PEUs changes, changes from one class type to another were characterized. Cross-tabulation analysis was performed to quantify PEUs changes from period 1 to period 2, from period 2 to period 3, and from period 1 to period 3. The PEUs changes in areas from one period to another period can be determined quantitatively and spatially by this analysis. The gains and losses experienced by various PEUs classes and analysis of the spatial trend of these changes for the study area were investigated for period 1, period 2, and period 3.

### 2.4.4. Selection of PEUs Transitions

Multiple mechanisms of transitions may occur in PEUs between two periods. All the PEUs changes and transitions were collected into a set of sub-models using the NN-MLP algorithm. The land cover change prediction was based on dependent and independent variables. In this work, six dependent variables (environmental factors), i.e., slope, aspect, DEM, distance from roads, temperature, and precipitation, were considered. These variables were selected according to the results of other studies in this field [8] (Dey, Nataraj Narayan, 2021) and according to the Kramer correlation coefficient. PEUs maps were considered independent variables. PEUs transition potential maps for each sub-model were produced based on PEUs transitions, as well as environmental factors with the help of an NN-MLP integrated in the LCM model.

### 2.4.5. Prediction of Future PEUs Changes

The MLP-MC model was used to predict and simulate the PEUs dynamics for a specified future date. Using the NN-MLP model, it is possible to determine the transitions that will be integrated for future PEUs change prediction. This model includes an input layer, many hidden layers, and an output layer, thereby known as a reputed change prediction model for its higher accuracy [34]. The MLP-MC model for the period 3 predicted map was created using the forecast model and the PEUs classification maps of period 1 and period 2. To validate the accuracy of the predicted map, we used the PEU classification map of the same period (period 3 classified map and period 3 predicted map). Finally, using the MLP-MC model, we obtained a PEU prediction map for the year 2036.

## 3. Results

### 3.1. Selecting the Optimal Time-Series Datasets

Figure 4a–c show the extracted NDVI temporal profiles for three periods. In these profiles, patterns and trends in NDVI changes can be observed for selected periods. The highest NDVI value belongs to the spring season, as this season is the peak of PEUs vegetative growth. In contrast, the lowest NDVI value can be observed in autumn and winter. As shown in Figure 4a–c, generally, the highest NDVI value occurs for selected periods between April and June. The multi-temporal images with the maximum NDVI values were selected for PEUs classification for each period (Table 1).

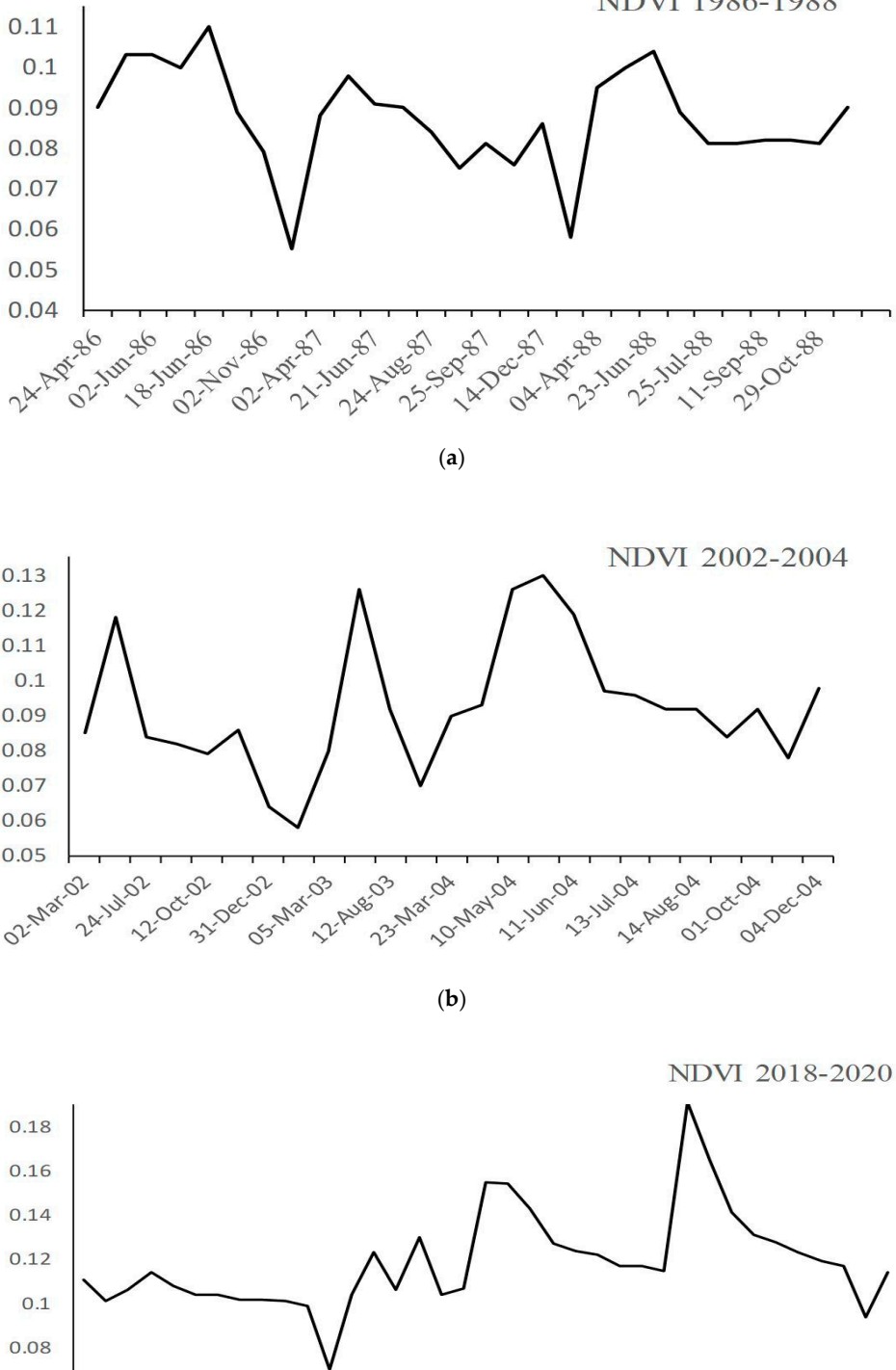

**Figure 4. (a)** The NDVI time-series profile of Landsat 5 TM (1986–1988), images for the periods of January to December in each year. (**b**) The NDVI time-series profile of Landsat 7 ETM+ (2002–2004), images for the periods of January to December in each year. (**c**) The NDVI time-series profile of Landsat 8 OLI (2018–2020), images for the periods of January to December in each year.

**Table 1.** Details of multi-temporal images used in this study for PEUs classification.

| Sensor | Year | Month/Day | Year | Month/Day | Year | Month/Day |
|---|---|---|---|---|---|---|
| Landsat 5 TM | 1986 | 24 April 17 May 2 June 18 June | 1987 | 2 April 4 May 20 May 5 June 21 June | 1988 | 4 April 7 June 23 June |
| Landsat 7 ETM+ | 2002 | 5 May 14 May 17 June | 2003 | 24 May | 2004 | 8 April 10 May 26 May 11 June 27 June |
| Landsat 8 OLI | 2018 | 25 May 10 June 26 June | 2019 | 26 April 28 May 13 June 29 June | 2020 | 12 April 28 April 14 May 30 May 15 June |

*3.2. PEUs Classification and Validation*

Using the RF classification algorithm, PEUs classification was performed for selected periods. Figure 5 shows three PEUs classification maps. The accuracy of the PEUs maps were was estimated using confusion matrices. In Table 2, the OA and OK of each period are presented. Also, the PA, UA, and KIA of PEUs are reported. The OA of the maps was 0.73, 0.77, and 0.81 for period 1, period 2, and period 3, respectively.

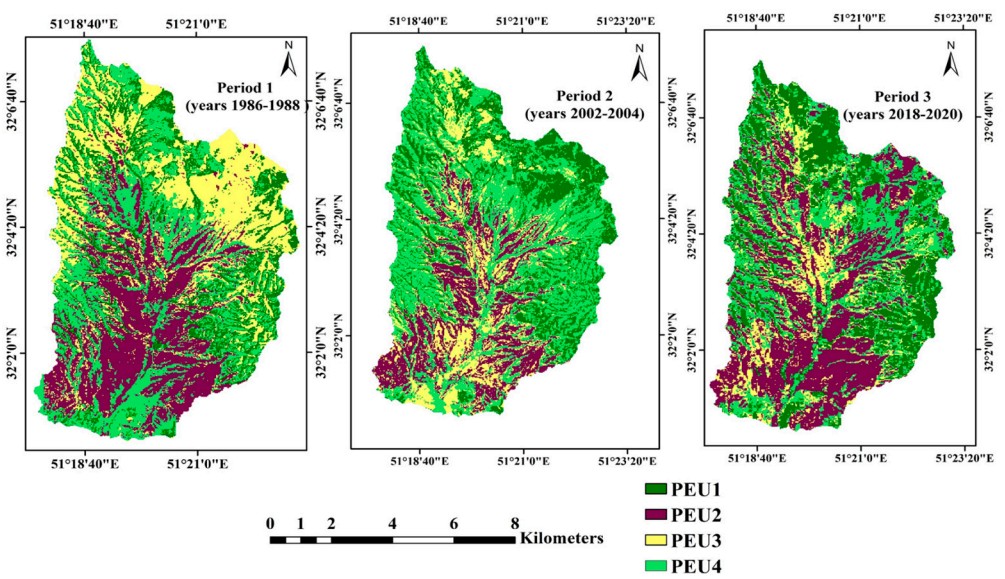

**Figure 5.** PEUs maps of period 1, period 2, and period 3.

**Table 2.** Confusion matrix results of PEUs classification of periods 1, 2, and 3.

| | Period | | | | | | | | |
|---|---|---|---|---|---|---|---|---|---|
| PEUs Class | Period 1 | | | Period 2 | | | Period 3 | | |
| | PA1 | UA1 | KIA1 | PA2 | UA2 | KIA2 | PA3 | UA3 | KIA3 |
| PEU1 | 0.84 | 0.74 | 0.84 | 0.91 | 0.77 | 0.87 | 0.91 | 0.91 | 0.88 |
| PEU2 | 0.79 | 0.64 | 0.50 | 0.84 | 0.72 | 0.76 | 0.75 | 0.82 | 0.67 |

**Table 2.** *Cont.*

| PEUs Class | Period 1 | | | Period 2 | | | Period 3 | | |
|---|---|---|---|---|---|---|---|---|---|
| | PA1 | UA1 | KIA1 | PA2 | UA2 | KIA2 | PA3 | UA3 | KIA3 |
| PEU3 | 0.58 | 0.75 | 0.63 | 0.67 | 0.89 | 0.58 | 0.84 | 0.67 | 0.75 |
| PEU4 | 0.70 | 0.78 | 0.56 | 0.67 | 0.73 | 0.56 | 0.75 | 0.90 | 0.68 |
| | OK = 0.63, OA = 0.73 | | | OK = 0.68, OA = 0.77 | | | OK = 0.74, OA = 0.81 | | |

PA: Producer's Accuracy %; UA: User's Accuracy %; and KIA: Kappa Index of Agreement %.

### 3.3. Change Analysis in PEUs Classes

As shown in Table 3(a), significant changes occurred in all PEUs over the three periods. These changes were analyzed from period 1 to period 2, period 2 to period 3, and period 1 to period 3 (Table 3(b)). PEU1 in period 1 covered an area of 1224.27 ha (15.82%) and it increased to 1582.12 ha (20.48%) and 1779.84 ha (23%) in period 2 and period 3, respectively. From period 1 to period 2, period 2 to period 3, and period 1 to period 3, PEU1 grew by 4.66%, 2.51%, and 7.18%, respectively.

**Table 3.** (a) Area of PEUs of period 1, period 2, and period 3. (b) Amount of changes in PEUs during period 1, period 2, and period 3 in the study area.

| (a) Area | | | | | | |
|---|---|---|---|---|---|---|
| | Period 1 | | Period 2 | | Period 3 | |
| PEU Class | Area (ha) | Area (%) | Area (ha) | Area (%) | Area (ha) | Area (%) |
| PEU1 | 1224.27 | 15.82 | 1582.12 | 20.48 | 1779.84 | 23 |
| PEU2 | 2089.62 | 27.08 | 1389.57 | 17.96 | 2567.79 | 33.19 |
| PEU3 | 2314.17 | 29.91 | 1444.58 | 18.67 | 1189.26 | 15.37 |
| PEU4 | 2108.7 | 27.25 | 3317.31 | 42.87 | 2199.69 | 28.43 |
| Total class | 7736.58 | 100 | 7736.58 | 100 | 7736.58 | 100 |
| (b) Amount of Changes | | | | | | |
| | Period 1–Period 2 | | Period 2–Period 3 | | Period 2–Period 3 | |
| PEU Class | Area (ha) | Area (%) | Area (ha) | Area (%) | Area (ha) | Area (%) |
| PEU1 | 360.95 | 4.66 | 194.72 | 2.51 | 555.67 | 7.18 |
| PEU2 | −699.97 | −9.04 | 1178.22 | 15.22 | 478.25 | 6.18 |
| PEU3 | −869.59 | −11.23 | −255.32 | −3.3 | −1124.91 | −14.54 |
| PEU4 | 1208.61 | 15.62 | −1117.62 | −14.4 | 90.99 | 1.17 |

PEU2 covered an area of 2089.62 ha (27.08%) in period 1, and it decreased to 1389.57 ha (17.96%) in period 2, while it increased to 2567.79 ha (33.19%) in period 3. From period 1 to period 2, PEU2 decreased by 9.04%, while from period 2 to period 3 and period 1 to period 3, PEU2 increased by 15.22% and 6.18%, respectively. Also, it was determined that in period 1, the PEU3 area covered an area of 2314.17 ha (29.91%) and it reduced to 1444.58 ha (18.67%) and 1189.26 ha (15.37%) in period 2 and period 3, respectively. A continuous decrease in PEU3 was observed by 11.23%, 3.30%, and 14.54% from period 1 to period 2, period 2 to period 3, and period 1 to period 3, respectively. The area covered by PEU4 in period 1 was 2108.7 ha (27.25%), and it increased to 3317.31 ha (42.87%) in period 2, while it decreased to 2199.69 ha (28.43%) in period 3. From period 1 to period 2, PEU4 increased by 15.62%, while during period 2 to period 3, PEU4 decreased by 14.4% and again increased by 1.17% from period 1 to period 3.

### 3.4. Transition Potential Modeling

### 3.4.1. Testing and Selecting Environmental Variables

This step identifies the environmental variables that have the ability to describe changes in the area. As shown in Figure 6, a total of six environmental variables, including the Digital Elevation Model (DEM), aspect, slope, distance from roads, precipitation, and temperature, were used for the transition potential modeling. These variables were selected according to other studies in this field [25,35], and also according to Cramer's V statistics. Cramer's V statistics were used to evaluate the potential power of these variables (as an independent variable) on the sub-model in which change was observed (as a dependent variable). The variables with a Cramer's V value of about 0.15 or higher are regarded as effective [36]. As presented in Table 4, the best Overall Cramer's coefficient of 0.44 was obtained using the DEM variable.

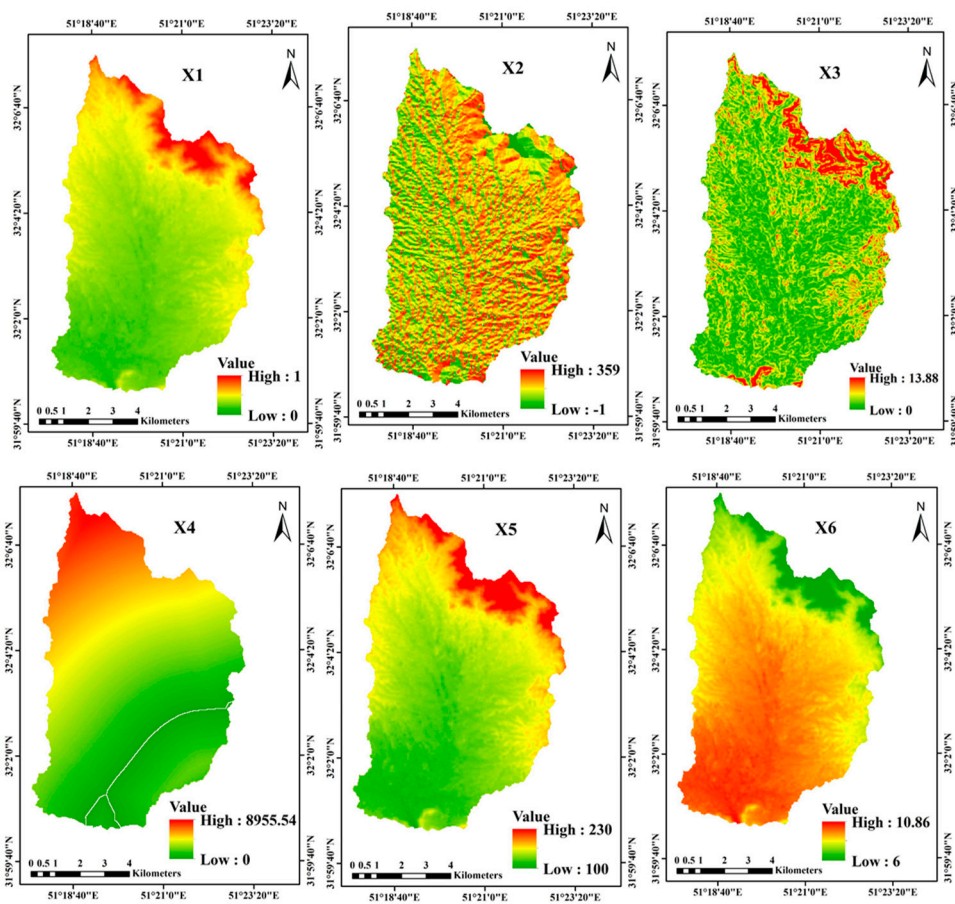

**Figure 6.** Environmental variable maps. (X1) DEM; (X2) aspect; (X3) slope; (X4) distance from roads; (X5) precipitation; (X6) temperature.

**Table 4.** Cramer's V statistics results for environmental variables.

| Environmental Variables | Overall Cramer's |
|---|---|
| Digital Elevation Model (DEM) | 0.4458 |
| Aspect | 0.182 |
| Slope | 0.194 |
| Distance from roads | 0.21 |
| Precipitation | 0.396 |
| temperature | 0.264 |

3.4.2. Transition Sub-Models

After reaching acceptable Cramer's V values for all environmental factors, subsequently, the transfer potential model was run. This model expresses the tendency of each image cell to receive a change from one PEU type to another type with respect to environmental variables. All the observed PEUs transitions were collected into a set of sub-models. Likewise, the transition potential was determined by developing twelve sub-models in the NN-MLP algorithm. Table 5 gives the results of transfer potential modeling using the NN-MLP algorithm. To evaluate the transfer potential modeling, three factors of Accuracy Rate, Testing RMS, and Training RMS were determined. The results in all sub-models show an accuracy of 61 to 89%.

**Table 5.** Results of accuracy assessing of transfer potential modeling using NN-MLP algorithm.

| Sub-Models | Testing RMS | Training RMS | Accuracy Rate |
|---|---|---|---|
| PEU1 to PEU2 | 0.27 | 0.27 | 89.14 |
| PEU1 to PEU3 | 0.39 | 0.39 | 78.16 |
| PEU1 to PEU4 | 0.43 | 0.43 | 71.51 |
| PEU2 to PEU1 | 0.41 | 0.43 | 73.26 |
| PEU2 to PEU3 | 0.42 | 0.42 | 75.27 |
| PEU2 to PEU4 | 0.4 | 0.39 | 77 |
| PEU3 to PEU1 | 0.41 | 0.4 | 76 |
| PEU3 to PEU2 | 0.39 | 0.4 | 79 |
| PEU3 to PEU4 | 0.44 | 0.44 | 70.53 |
| PEU4 to PEU1 | 0.37 | 0.37 | 82 |
| PEU4 to PEU2 | 0.41 | 0.4 | 75.29 |
| PEU4 to PEU3 | 0.47 | 0.48 | 61.18 |

*3.5. Prediction and Validation of PEUs Changes with Markov Chain Modeling*

To predict a PEU map for period 3 (years 2018–2020), two different PEUs maps of period 1 and period 2 were used to create the prediction map (Figure 7) and transition probability matrix (Table 6) using the MC prediction process. As shown in Table 6, the probability of a change of PEU4 into PEU1 and PEU2 in period 3 from period 1 to period 2 is 33.91% and 17.22%, respectively. Also, the probability that PEU3 converts into PEU2 in period 3 is 26.99%.

**Table 6.** Markov Chain transition probability matrix of changes among PEUs (period 1–period 2) for period 3.

| PEU Class | PEU1 | PEU2 | PEU3 | PEU4 |
|---|---|---|---|---|
| PEU1 | 0.7563 | 0 | 0.1149 | 0.1228 |
| PEU2 | 0 | 0.7958 | 0.1282 | 0.0761 |
| PEU3 | 0.028 | 0.2699 | 0.5162 | 0.1853 |
| PEU4 | 0.3391 | 0.1722 | 0.1411 | 0.3476 |

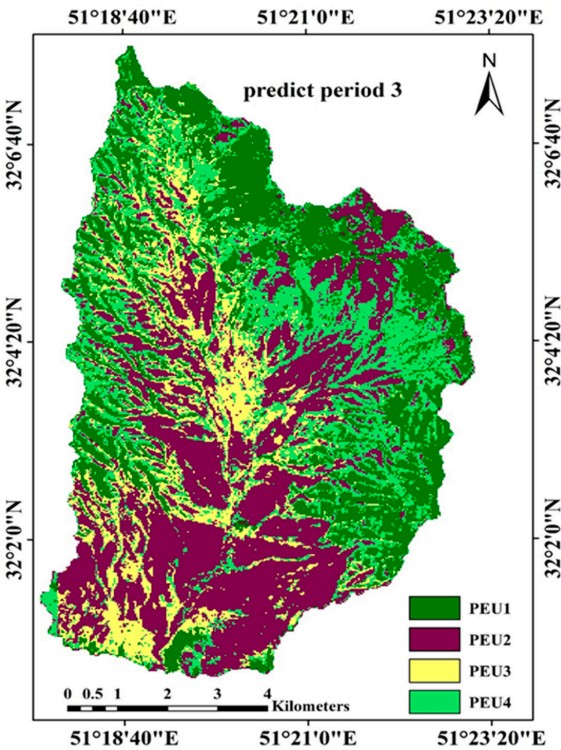

**Figure 7.** Predicted PEUs map for period 3 (years 2018–2020).

To evaluate and validate the MC prediction process, the period 3 classified map and the period 3 predicted map were used. Cross-tabulation analysis shows that the Overall Cramer's value is 68.53% and the Overall Kappa is 77.6%.

In addition, Figure 8 shows the side-by-side comparison of PEUs in two classified and predicted maps of period 3. In this map, the PEUs that are classified in both maps in the correct pixels are marked with 1-1, 2-2, 3-3, and 4-4. This means that there are pixels that are classified in the same class in both maps. The parts that are white on the map show the wrong pixels. This means that this model has incorrectly predicted the white parts.

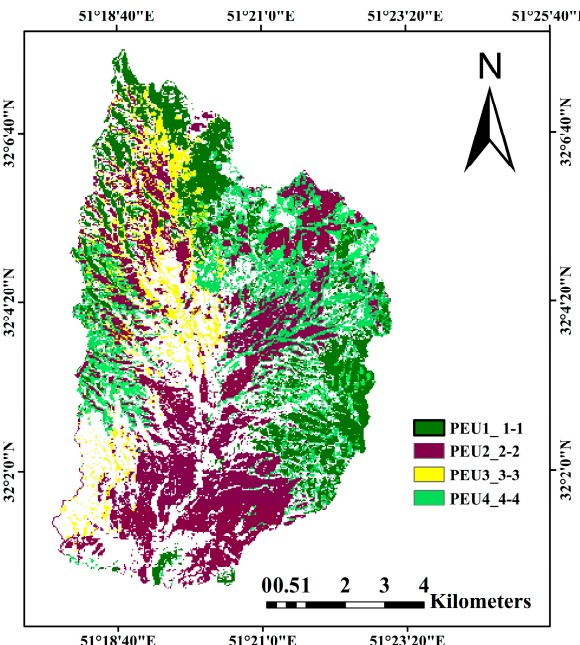

**Figure 8.** The map resulting from the comparison of the classified and predicted maps of period 3 (years 2018–2020).

### 3.6. Prediction of Future PEUs Changes

The PEUs maps of period 2 and period 3 were used to predict the future PEU scenarios for the year 2036 using the MLP-MC model. The predicted PEUs map (Figure 9) and transition probability matrices (Table 7) were produced using the PEUs maps of period 2 and period 3. By utilizing the PEU of period 3 as the base map, the transition potential maps, and the transition probability matrices of period 2 to period 3, the future PEU scenarios were predicted for the year 2036. The probability of PEU4 turning into PEU1 and PEU2 in the year 2036 is 32.10% and 15.88%, respectively. Also, the probability of PEU3 turning into PEU2 and PEU4 in 2036 is 19.17% and 20.93%, respectively.

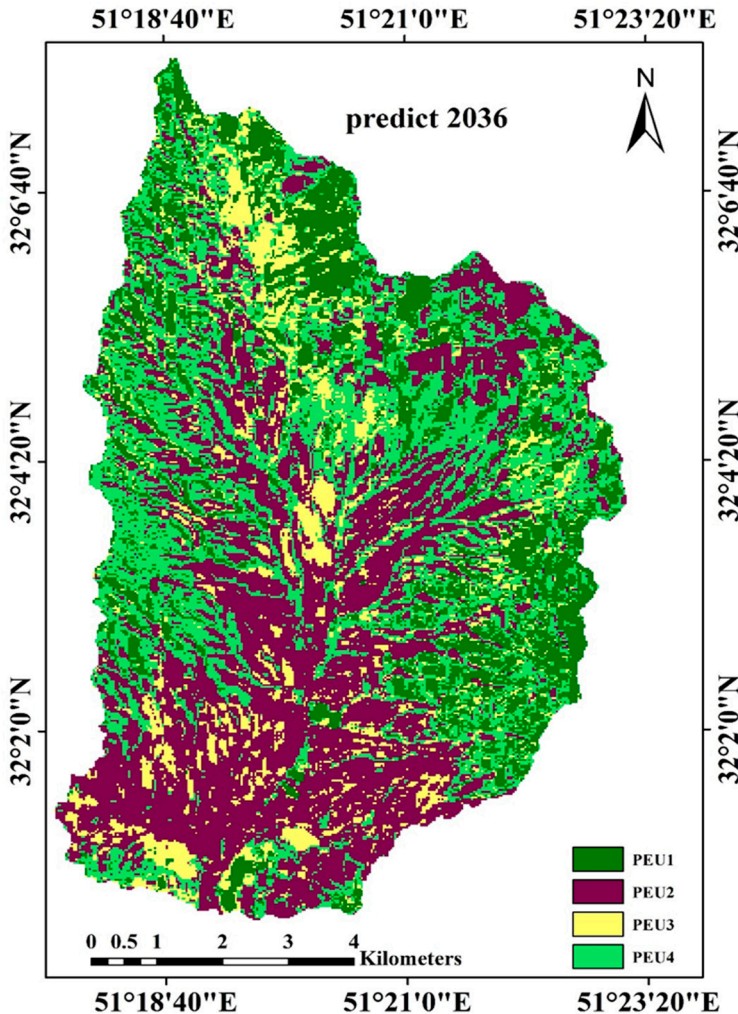

**Figure 9.** Predicted PEUs map for year 2036.

**Table 7.** Markov Chain transition probabilities matrix of changing among PEUs (period 2–period 3) for 2036.

| PEU Class | PEU1 | PEU2 | PEU3 | PEU4 |
|---|---|---|---|---|
| PEU1 | 0.7437 | 0 | 0.0935 | 0.1628 |
| PEU2 | 0 | 0.8369 | 0.1482 | 0.0150 |
| PEU3 | 0 | 0.1917 | 0.5990 | 0.2093 |
| PEU4 | 0.3210 | 0.1588 | 0.1119 | 0.4084 |

The areas of the PEUs classes of the predicted map of 2036 are presented in Table 8. A comparison of the prediction map of the year 2036 and the classification map of period 3 (2018–2020) reveals that PEU2 and PEU4 will increase by 333.63 ha (4.30%) and 60.46 ha

(0.07%), respectively. Meanwhile, PEU1 and PEU3 will decrease by 153.07 ha (1.98%) and 241.02 ha (3.12%), respectively.

**Table 8.** Area distribution of PEUs classes for predicted year 2036.

|  | Area (ha) | Area % |
|---|---|---|
| PEU1 | 1626.77 | 21.02 |
| PEU2 | 2901.42 | 37.50 |
| PEU3 | 948.24 | 12.25 |
| PEU4 | 2260.15 | 29.21 |
| Total area | 7736.58 | 100 |

## 4. Discussion

Knowledge and understanding of historical PEUs patterns, changes, and future trends are important in proactive environmental management. To enable the study and prediction of long-term PEUs dynamics, it is critical to analyze and understand the various changes shaping the land cover, which, among other factors, can help to minimize undesired changes in plant communities [6]. PEUs change is the result of complex interactions between human activities and environmental factors from the past to the present [33]. We started with the four dominant PEU classes in the study area: PEU1 (*As ve*), PEU2 (*Br to*), PEU3 (*Sc or*), and PEU4 (*As ve—Br to*).

With the purpose of understanding the spatial–temporal dynamics of PEUs, we divided this work into four sections: (1) Preparation of PEU multi-temporal classification maps and accuracy assessment for three different periods. (2) Change analysis of periods 1–2, periods 2–3, and periods 1–3. (3) Prediction of period 3 by the MLP-MC model, and comparison of the prediction results with the classified PEU map of period 3. (4) Prediction of future scenarios of PEUs for the year 2036 using the MLP-MC model that produced the best results in the prediction of period 3. These sections are further discussed below.

### 4.1. PEUs Multi-Temporal Classification Maps for Three Periods

This study used time series of Landsat data in three periods of three years that include period 1 (Landsat 5 TM images for 1986–1988), period 2 (Landsat 7 ETM+ images for 2002–2004), and period 3 (Landsat 8 OLI images for 2018–2020). According to the NDVI temporal profile, we selected the optimal multi-temporal images for each period. As shown in Figure 5, the most informative temporal windows were observed in spring for the period of April through June. A total of 12 cloudless images were extracted for period 1, 9 cloudless images were extracted for period 2, and 12 cloudless images were extracted for period 3 (Table 1).

The GEE platform allows the synchronization of all the Landsat data and then the creation of a high-quality multi-temporal dataset using codes already provided [37]. The RF algorithm was chosen for PEUs classification. The RF algorithm is a tree-based machine learning method that leverages the power of multiple decision trees for making decisions and is suitable for situations when we have a large dataset [1]. At this stage, three multi-temporal PEU maps were obtained with an OA of 73%, 77%, and 81% for period 1, period 2, and period 3, respectively. The increasing OA from period 1 to period 3 can be related to the development of Landsat generations' capabilities specifically in terms of radiometric properties and the consequently decrease in the noise-to-signal ratio. PEUs as a sub-class of rangeland cover are involved. Sub-classes of vegetation cover are more similar in terms of their spectral reflectance than that of a higher hierarchical land cover classification [7]. Thereby, PEUs classification process using an optimal time-series dataset is required to accurately identify and discriminate the past and current trends as well as to predict future trends of PEUs.

### 4.2. Analysis of PEUs Changes

The study area during period 1 (1986–1988), period 2 (2002–2004), and period 3 (2018–2020) experienced intense changes in PEUs. There are significant changes that occurred in all PEUs classes. Each of the PEU represents the different succession stages of each period in the study area. Natural disturbances and human activities such as fire, drought, land tillage, and irregular grazing can change the plant community's successional stages. Thereby, it will lead to a decrease in vegetation and a descending trend in plant community succession. The PEUs classification maps of period 1, period 2, and period 3 demonstrate PEUs changes well (Figure 5). In period 1, PEU3 covered most of the study area and was the dominant PEU with 2314.17 ha (29.91%). PEU3 is dominated by semi-shrub species (*Sc or*). This plant species is ecologically an invasive species, and in areas where the plant community has been destroyed by tilling, it becomes dominant in the area [38]. Examination of 1986-1988 documents shows that this area was heavily tilled and cultivated but was abandoned due to low yields. Thereby, PEU3 became the dominant PEU in the area, which shows plant succession's declining trend and the severe destruction of the plant community. Meanwhile, due to the implementation of appropriate management practices in the region, such as tillage inhibition, PEU3 reduced to 18.67% and 15.37% in period 2 and period 3, respectively. Thereby, after 16 years (period 2) and 32 years (period 3), much of the area has recovered its vegetation with a suitable canopy cover, whereby shrubs (PEU1) and perennial grasses (PEU2) became dominant. Likewise, the plant succession trend is upward and positive. In period 2, PEU2 had the lowest coverage in the study area, with 1389.57 ha (17.96%). Meanwhile, shrubs and non-palatable species (e.g., PEU1) have become more widespread. PEU2 is dominated by grass species (*Br to*). These plant species are palatable to livestock and are destroyed by grazing before their growth is completed [38]. Examination of 2002–2004 documents shows that this area was affected by severe grazing. Therefore, PEU2 decreased in period 2. After 16 years, PEU2 increased to 33% in period 3, mainly due to appropriate management practices and preventing severe grazing. Much of the area is dominated by perennial grasses (i.e., PEU2). This suggests that, with the implementation of proper management policies and the prevention of destructive processes such as tilling and heavy grazing, the plant community will approach a climax stage.

### 4.3. Transition Probabilities of Change among PEUs

For modeling future PEUs dynamics, the transition probability assessment among different PEUs is an important aspect. The NN-MLP algorithm was used to determine the weights of transitions for two periods that were included in the transition probability matrix using the MC model for future prediction. The MC model is very effective in determining the behaviors and extent of land cover changes by analyzing land cover maps of two periods [8]. Based on a projection of the transition potentials, the method determines exactly how many PEUs would transition from the specified date to the predicted date in the future [33]. In this study, in the first step, two different PEUs maps of period 1 and period 2 were used to create the transition potential maps for period 3 (Table 6). For validation of the MC prediction process, the PEUs prediction map of period 3 was compared with the PEUs classification map of period 3 using cross-tabulation analysis. It was found that the Kappa Index of Agreement value is 77.6%, showing good agreement between the observed and predicted PEUs maps. Then, in the next step, the PEUs map of period 2 and period 3 were used to predict the future PEUs changes for 2036 (Table 7). In both stages, the probability of the change of PEU4 into PEU1 and PEU2 in period 3 and 2036. Also, the probability of change of PEU3 into PEU2 is significant in both predicted dates. This suggests that the MC is a robust model and can predict future PEUs changes.

### 4.4. Prediction of Future PEUs for Year 2036

The last step is the prediction of future scenarios of PEUs for the year 2036 using the MLP-MC model that produced the best results in the prediction of period 3. By utilizing

the PEU of period 3 as the base map, the transition potential maps, and the transition probability matrices of period 2 to period 3, the future PEU scenarios were predicted for 2036. The prediction results for 2036 are shown in Table 8; the dominant vegetation of the area for 2036 will be PEU2 and PEU4 with 37.50% and 29.21%, respectively. The predicted results further show that by 2036, PEU3 will have the lowest coverage with 12.25%. Comparing the 2036 predicted map with the period 3 classified map shows that the PEU1 area decreases (153.07 ha). PEU1 is dominated by shrub species (*As ve*). Field investigations in recent years have shown that tragacanth gum extraction from *As ve* has dramatically increased. However, little information is available about the best method of gum harvesting, which is economically efficient and maintains the health of the plant. This study's results demonstrate that the MLP-MC method has the ability to predict and describe future PEU changes that are the result of current activities. Also, by aiming to compare three prediction models, Cellular Automata–Markov Chain (CA-MC), Stochastic Markov Chain (ST-MC), and MLP-MC, to predict land cover changes in the Varanasi district, ref. [25] reported that the MLP-MC prediction model had the best results for an accurate understanding of changes and predicting future landscape scenarios. The implementation of an arsenal of management activities and natural resource conservation policies will reduce pressure on plant communities such as PEUs, thereby contributing to natural environmental sustainability even in the future.

## 5. Conclusions

This study aimed to reveal and analyze the dynamics in PEUs patterns. Nowadays, land use change patterns and urban area development and construction are considered the main variables in the analysis and monitoring of the terrestrial environment and natural system changes. In many areas, however, land cover and plant communities have declined for multiple reasons. The main cause is human activities, such as intense grazing, fire, tilling, and drought. But what is often unsaid is that anthropogenic processes and management activities are the most important drivers in plant community changes. It is therefore essential to identify these activities for the sake of future planning and natural resource management. This study evaluated the MLP-MC model to identify the main processes in PEUs changes from the past to the future and predicted future PEUs dynamics. We first selected three periods with 16-year time intervals. PEUs information extracted for period 1 and period 2 was employed to predict PEUs changes for period 3 using the MLP-MC prediction model. The predicted results were compared with the classified PEUs information for period 3 to appraise the validity of models through cross-tabulation analysis and kappa index statistics (Overall Kappa: 77.6%). Finally, we predicted future PEUs changes 16 years ahead of period 3, for the year 2036. The results demonstrated that the MLP-MC model is efficient in predicting PEUs patterns in the future. The basis for this model is the anthropogenic and natural processes of the past. Our study suggests that the impact of anthropogenic processes and management activities will become visible in the natural environment and ecosystem in less than a decade.

**Author Contributions:** Methodology, A.E. and M.A.; conceptualization, A.E. and M.A.; software, M.A.; validation, A.E.; investigation, M.A.; resources, M.A.; formal analysis, M.A.; data curation, A.E. and M.A.; writing—original draft preparation, M.A. and J.V.; writing—review and editing, A.E., J.V., A.A.N. and E.A.; supervision, A.E.; visualization, M.A. and A.E.; project administration, A.E.; funding acquisition: J.V. All authors have read and agreed to the published version of the manuscript.

**Funding:** This work was supported by Shahrekord University. J.V. was funded by the European Union (ERC, FLEXINEL, 101086622). Views and opinions expressed are however those of the author(s) only and do not necessarily reflect those of the European Union or the European Research Council. Neither the European Union nor the granting authority can be held responsible for them.

**Data Availability Statement:** Data supporting this study may be provided by the first author upon a reasonable request. The data are not publicly available due to the privacy of parts of the data.

**Conflicts of Interest:** The authors declare no conflict of interest.

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
