# Peer review of "Monitoring of Plant Ecological Units Cover Dynamics in a Semiarid Landscape from Past to Future Using Multi-Layer Perceptron and Markov Chain Model"

_remotesensing, doi:10.3390/rs16091612_

Round 1

Reviewer 1 Report

Comments and Suggestions for Authors

Thank you for inviting me to review. The paper utilizes historical data trends to forecast future scenarios, which contributes positively to the development and protection of local ecological environments. I have some suggestions regarding the content:

1.Figure 3’s flowchart should be redrawn with a revised layout.

2.Lines 111-115 should elaborate on the rationale behind the PEU segmentation. Why are there four PEUs? The theoretical basis for this segmentation should be elucidated.

3.The division of the dataset into a 6:4 ratio for the validation and training sets, as mentioned in lines 154-155, is unusual. Typically, the training set should have more samples than the validation set to provide a solid foundation for model learning. Common practice is to allocate 70% or 80% of the data for training, with the remaining 30% or 20% for validation. In your study, assigning the majority of data (180 samples) for validation with only 120 samples for training could result in inadequate model training. The model may not learn the full complexity and variability of the data on a smaller dataset, affecting its generalizability. Please justify the scientific rationale for this distribution.

4.In lines 130-142, the variation in hydrothermal conditions across different seasons results in different vegetation index values. Is the GEE-derived data for LANDSAT5, LANDSAT7, and LANDSAT8 representative of the same time period each year, or are they annual averages? Please clarify the data specifics.

5.Figure 4 should be reorganized to improve the numbering and visual presentation of the chart.

6.The hyperparameter optimization for the Random Forest and MLP-MC models should be detailed, explaining how model optimality is ensured.

7.In English technical writing, terminology should be clearly defined or described beforehand, as seen with 'ST-MC' and 'CA-MC' mentioned in line 477.

Comments on the Quality of English Language

The overall proficiency in English is commendable. However, attention should be given to the segmentation of long and short sentences and the articulation of technical terms.

Reviewer 2 Report

Comments and Suggestions for Authors

It is very interesting and valuable for monitoring and predicting the PEUs in hostoric, current and futural dates with MLP-MC modesl, especially in arid and semiarid landscape. The methods, results and discussion are written well, specifical comments need pay attention. 

1.In title of 3.5 section, it will be better using "with" to replace "in".

2. In L316 of 3.5 section, the 2036 should be period 3. 

3.In 3.5 section, after the validation between period 3 classified map and the predicted map, namely the Overall Cramer's value is 68.53% and the Overall Kappa is 77.6%, how about the accuracy result compared with former similar literatures?  the accuracy is not very high and satisfied in fact, what is the reason and the research limit? so as to improve the accuracy, it should be discussed or issued in next future work. 

Reviewer 3 Report

Comments and Suggestions for Authors

I reviewed the manuscript entitled "Monitoring of PEUs cover dynamics in a semiarid landscape from past to future using Multi-layer perception and Markov chain model" by Aghababaei et al. The authors used Landsat Archive images to generate PEUs maps in three intervals to analyze the spatio-temporal changes. Later, the generated maps were used to predict the future PEUs of the study area using the MLP-MC approach. The manuscript's topic is interesting and falls within the scope of the Journal. I have provided my comments below.

1- I recommend that the authors reread the manuscript and improve its English quality, as a few sentences are difficult to understand and require modifications.

2- Lines 154-155: How did you divide the samples into training and test samples? I assume a random sampling strategy was adopted. In this case, did you check the distances between training and test samples to reduce/eliminate the spatial autocorrelation between them? Spatial autocorrelation can bias the accuracy assessment results.

3- I suggest authors to replace Figure 3. It would be better to place it at the beginning of Section 2.4, with complementary explanations. In this case, the readers can obtain a suitable overview of the implemented approach and make other parts easy to follow.

4- Lines 225-226; As clear in NDVI profiles (Figure 4), there is only one maximum NDVI value in each period. So, please clarify the sentence "The multi-temporal images with the maximum NDVI values were selected for PEUs classification for each period (Table 1)". It seems that you have selected time-series images with NDVI values over the top percentile. Please clarify.

5- There is one issue regarding the time-series selection. Why was the high NDVI value, along the NDVI profile, considered as a measure? Please justify. Moreover, how were the NDVI profiles for the region generated? Why not present the time-series NDVI behavior of each PEUs and then select the time-series with the highest NDVI differences between the PEUs? This would allow for obtaining higher classification accuracy, as the input data can distinguish the PEUs more efficiently. The current selection strategy lacks theoretical and practical backgrounds and justifications.

6- Please include the confusion matrixes of the classification maps.

7- Please state the date of the fieldwork for sample collections. I assume that the samples were collected in the latest year through fieldwork. However, the same test samples seem to have been used for the validation step of the PEUs maps. How did you ensure that the samples were not changed during the time? Did you consider any reference samples migration method to exclude the change pixels? Please provide detailed explanations in this regard, as it is necessary to make the evaluations robust.

8- Please compare the predicted and generated PEUs maps in 2020 in the spatial domain and provide the results as a new Figure. This will allow for understanding the spatial performance/limitations of the prediction model.

9- As I understand, the PEUs map in 2036 was predicted using PEUs maps in 1998 and 2004. Why not add the generated PEUs map of 2020 to enhance the prediction capability of the model?

Comments on the Quality of English Language

The writing style, in general, is good, though in some cases, it is hard to understand and follow the objective of sentences. I recommend that authors re-read the manuscript and improve its English quality.

Round 2

Reviewer 3 Report

Comments and Suggestions for Authors

I appreciate the authors' efforts in addressing almost all comments. However, the response to comment 8 was not convincing and thus needed further investigation. As I suggested, the authors should compare the predicted and exact (classification results) PEUs maps in 2020 in the spatial domain across the study area. A new figure is required to show the spatial inconsistency (similar to a change map) between the predicted and exact PEUs maps. This will allow to obtain a spatial understanding of the prediction model.
